# Large-scale image-based profiling of single-cell phenotypes in arrayed CRISPR-Cas9 gene perturbation screens

Reinoud de Groot[1] iD, Joel Lüthi[1,2] iD, Helen Lindsay[1] iD, René Holtackers[1] & Lucas Pelkmans[1,*] iD

## Abstract

High-content imaging using automated microscopy and computer vision allows multivariate profiling of single-cell phenotypes. Here, we present methods for the application of the CISPR-Cas9 system in large-scale, image-based, gene perturbation experiments. We show that CRISPR-Cas9-mediated gene perturbation can be achieved in human tissue culture cells in a timeframe that is compatible with image-based phenotyping. We developed a pipeline to construct a large-scale arrayed library of 2,281 sequence-verified CRISPR-Cas9 targeting plasmids and profiled this library for genes affecting cellular morphology and the subcellular localization of components of the nuclear pore complex (NPC). We conceived a machine-learning method that harnesses genetic heterogeneity to score gene perturbations and identify phenotypically perturbed cells for in-depth characterization of gene perturbation effects. This approach enables genome-scale image-based multivariate gene perturbation profiling using CRISPR-Cas9.

**Keywords** arrayed library; CRISPR-Cas9; functional genomics; nuclear pore complex; single-cell phenotypic profiling

**Subject Categories** Chromatin, Epigenetics, Genomics & Functional Genomics; Genome-Scale & Integrative Biology; Methods & Resources

**Mol Syst Biol. (2018) 14: e8064**

## Introduction

Forward and reverse genetic screens in mammalian cells and model organisms have provided a wealth of information about gene function (Boutros & Ahringer, 2008; Liberali et al, 2015). Nonetheless, the role of a significant proportion of genes remains unknown and additional gene functions remain to be elucidated. The discovery of the CRISPR-Cas system has revolutionized functional genetic screening because, unlike RNAi, CRISPR-Cas9 targets genes at the DNA level and can therefore generate genetic null alleles, resulting in complete genetic perturbation effects. For this reason, CRISPR-Cas9 has been used in large-scale functional genomic screens (Shalem et al, 2015). Most screens performed to date employ a pooled screening strategy, which can identify genes that cause differential growth in screening conditions (Koike-Yusa et al, 2013; Shalem et al, 2014; Wang et al, 2014). However, pooled screening precludes multivariate profiling of single-cell phenotypes. This can be partially overcome by combining pooled screening with single-cell RNA-seq, but this does not easily scale to the profiling of thousands of single cells from thousands of perturbations, and is limited to features that can be read from RNA transcript profiles (Adamson et al, 2016; Dixit et al, 2016; Jaitin et al, 2016; Datlinger et al, 2017). Moreover, sequencing-based approaches do not provide information on cellular size or morphology, cellular microenvironment, or on the subcellular organization of intracellular structures such as the nuclear pore complex (NPC). Image-based phenotyping using automated microscopy is ideally suited to study such phenotypes. Recently, methods to perturb cells in a pooled format, followed by image-based phenotyping and in situ genotyping were developed for prokaryotic model systems (Emanuel et al, 2017; Lawson et al, 2017). An alternative screening strategy involves seeding cells in multi-well plates that contain reagents that perturb one specific gene per well. This arrayed screening strategy allows detailed, image-based phenotyping of populations of cells in which specific genes are perturbed (Boutros et al, 2015; Liberali et al, 2015; Caicedo et al, 2016). Recently, a number of studies have applied the CRISPR-Cas9 system to an arrayed format, but these were limited in scale and only obtained well-averaged readouts with low information content (Hultquist et al, 2016; Tan & Martin, 2016; Strezoska et al, 2017), not realizing the full potential that image-based multivariate single-cell phenotypic profiling could bring. Importantly, CRISPR-Cas9 is not 100% effective in all targeted cells, which can be the result of in-frame repair of the CRISPR-Cas9-induced DNA lesions, a failure to target all functional alleles or limited efficacy of the CRISPR-Cas9 system (Shalem et al, 2015). We present an approach to address this problem, allowing us for the first time to combine the power of CRISPR-Cas9 with high-content, image-based profiling of single-cell phenotypes across thousands of genetic perturbations.

1   Institute of Molecular Life Sciences, University of Zürich, Zürich, Switzerland
2   Systems Biology PhD program, Life Science Zürich Graduate School, ETH Zürich and University of Zürich, Zürich, Switzerland
    *Corresponding author. Tel: +41 44 63 53 123; E-mail: lucas.pelkmans@imls.uzh.ch

# Results and Discussion

We devised an experimental strategy for the application of the CRISPR-Cas9 system in an arrayed screening format. To allow maximum flexibility with regard to the cell line and assay used for screening, we opted for a one-component system where the coding sequence for SpCas9, a chimeric gRNA and a fluorescent protein (tdTomato) is combined on a single plasmid. We introduced targeting plasmids into human tissue culture cells by reverse transfection and assayed expression of the targeted gene by quantitative immunofluorescence (Fig 1A). As a proof of concept, we targeted the transferrin receptor (*TFRC*) in HeLa cells and assessed TFRC expression in approximately 4,000 single cells per experimental condition. A subpopulation of cells (which expresses tdTomato) loses TFRC expression starting 2 days post-transfection (Fig 1B and C), indicating that these cells are functionally genetically perturbed. The proportion of genetically perturbed cells increased at longer times after transfection. We also targeted the genes *LAMP1* and *YAP1* in HeLa cells and additionally show that the approach is effective in U2OS cells (Figs 1D and EV1A, B and C).

To systematically test our approach across multiple genes, we automated the selection of gRNA sequences with high predicted on-target efficacy (Doench *et al*, 2014). We selected gRNA sequences to target separate, expressed exons, while avoiding the first or last exons of transcripts (Fig EV1D). We employed a single-molecule fluorescence *in situ* hybridization (smFISH) technique (Battich *et al*, 2013) to detect the cells in which transcripts are depleted due to nonsense-mediated decay, which results from CRISPR-Cas9-induced frameshift mutations (Fig 1E, F and G). We targeted 26 genes with three targeting plasmids each. 72% of the targeting plasmids perturbed gene expression in more than 30% of transfected cells, indicating that we can reliably select functional gRNA sequences (Fig 1H, Table EV1).

We subsequently developed a cost-effective pipeline to produce a large-scale, arrayed library of sequence-verified CRISPR-Cas9 targeting plasmids. As a proof of principle, we constructed a library consisting of 2,281 transfection-grade plasmid preparations targeting 1,457 genes that are annotated with gene ontology (GO) terms of various post-translational modifications (Fig EV2, Dataset EV1). We transfected HeLa cells with the plasmids in 384-well plates, stained DNA and total protein and subjected the cells to immunofluorescence with mAb414, a monoclonal antibody that binds phenylalanine–glycine (FG) repeats present in several subunits of the nuclear pore complex (NPC; Davis & Blobel, 1986). We stained for this marker because the regulation of NPC assembly in interphase is incompletely understood (Otsuka *et al*, 2016; Weberruss & Antonin, 2016) and the subcellular localization of NPC components can only be investigated using microscopy. We imaged approximately 4,000 cells per targeted cell population and extracted a multivariate set of features describing the size and shape of the cells and intensity and texture of the fluorescent markers in specified subregions of every cell (Stoeger *et al*, 2015; Fig EV3A and B).

Our experimental approach generates transfected T(+) cells, which may be genetically perturbed, and non-transfected T(−) cells, which are genetically wild-type. We leveraged this aspect to address two challenges in the analysis of large-scale image-based profiling experiments; technical well-to-well variation and the identification of significant perturbation effects in high-dimensional single-cell datasets (Loo *et al*, 2007; Liberali *et al*, 2015; Caicedo *et al*, 2016). First, we used the T(−) cells as in-well controls to standardize all single-cell features and correct for technical variability between wells. Second, we trained logistic regression classifiers (Friedman *et al*, 2010) to attempt to categorize T(+) and T(−) cells from the same well based on a set of single-cell features (Fig 2A, Tables EV2 and EV3) and calculated a classification score based on the accuracy of the classifier. This approach takes the full heterogeneity among both wild-type and perturbed cells into account and thus addresses a major limitation of well-averaged approaches.

We observed that not every T(+) cell is phenotypically perturbed (Fig 1C, D, G and H), which complicates the analysis of gene perturbation effects. To address this issue, we used the classifiers that we fitted to the targeted cell population to calculate the predicted value (PV) for every individual cell. Cells with a positive PV are classified in the phenotypically perturbed class and a negative value indicates classification in the wild-type class. By limiting our analysis to T(+) cells with a high positive PV value, we discard the T(+) cells that are phenotypically wild-type. To illustrate this point, we targeted *NUP160,* which causes a strong phenotypic effect in single cells. Here, many cells have a high PV, which are almost exclusively T(+) cells (Fig 2C). In contrast, cells transfected with a control plasmid have a low absolute PV because T(+) and T(−) cells are indistinguishable in multivariate feature space (Fig 2B). We colour-coded cells from the *NUP160* targeted population for the expression of the tdTomato marker and PV. T(−) cells display the wild-type mAb414

---

**Figure 1. CRISPR-Cas9-mediated gene perturbation by transient transfection of targeting plasmids.**

A   Schematic overview of CRISPR-Cas9-mediated gene perturbation by transient transfection of a targeting plasmid. tdTomato expression (magenta) marks transfected cells. Single-cell measurements are obtained by quantitative immunofluorescence (green) combined with computer vision and automated cell segmentation, see text for details.

B   tdTomato (magenta) and TFRC (green) expression in HeLa cells transfected with a control plasmid, or a *TFRC* targeting plasmid. Scale bar, 50 μm.

C   Quantification of normalized TFRC staining per cell, 1–4 days after transfection of a *TFRC* targeting plasmid. Violin plots of normalized TFRC staining intensity in all analysed cells (grey) or tdTomato expressing (T(+), magenta) cells.

D   Quantification of the efficacy of genetic perturbation by *TFRC*, *LAMP1* and *YAP1* targeting plasmids; bars indicate the percentage of genetically perturbed T(+) cells. The mean ± standard deviation of three independent experiments is displayed.

E   Evaluation of genetic perturbations in single cells using bDNA FISH. Schematic representation of the expected phenotype in wild-type and functionally genetically perturbed cells.

F   bDNA FISH staining of *TFRC* mRNA in HeLa cells transfected with a control plasmid, or a *TFRC* targeting plasmid. Cell outlines are indicated and colour-coded white for T(−) cells, magenta for T(+) cells. Scale bar, 50 μm.

G   Quantification *TFRC* mRNA spots in cells transfected with a control plasmid, or a *TFRC* targeting plasmid. Violin plots of *TFRC* mRNA spot counts per T(+) cell.

H   Heatmap representation of the efficacy of targeting plasmids designed to perturb 26 selected genes as assayed by smFISH.

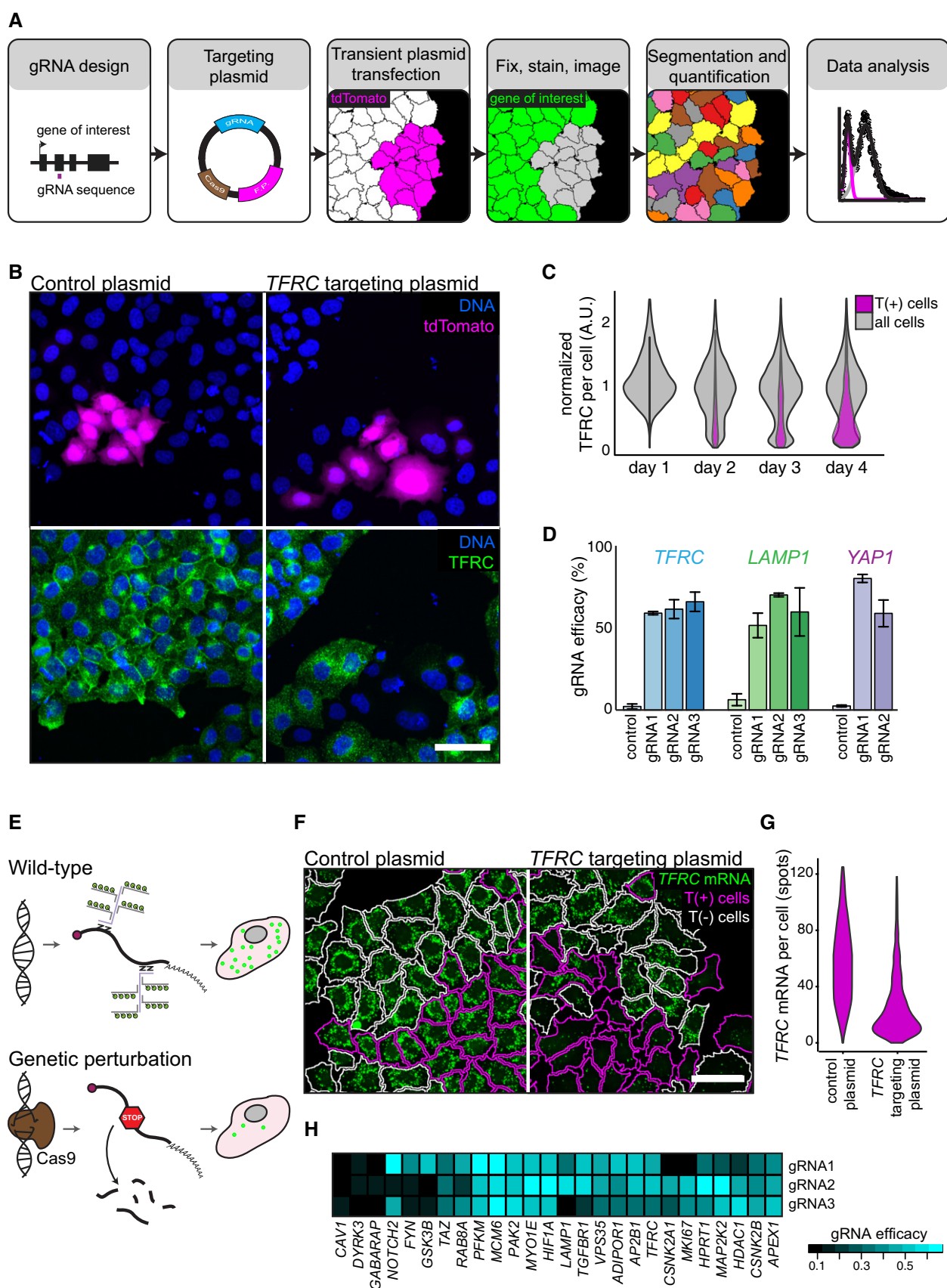

**Figure 1.**

**A**

1. Mosaic population of wild-type and genetically perturbed cells

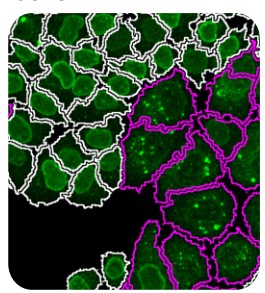

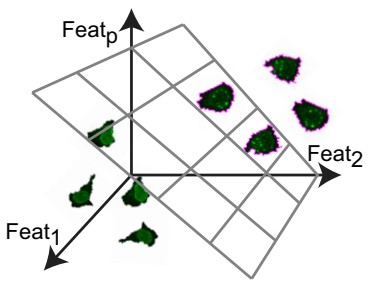

2. Standardize single cell features to T(-) cells

$$Cell_x\ Feat_y' = \frac{Cell_x\ Feat_y - \mu(Feat_y(T\text{-}))}{\sigma\ (Feat_y(T\text{-}))}$$

3. Logistic regression model to classify wild-type and transfected cells

$$\ln\left(\frac{F(x)}{1-F(x)}\right) = \beta_0 + \beta_1 X_1 + \beta_2 X_2 + ... + \beta_p X_p$$

Misclassification error
Single cell prediction

**B**

control

predicted value

**C**

*NUP160*

predicted value

T(-) ●
T(+) ●

**D**

*NUP160* targeting plasmid

missegmented
T(+)(P.V.>0.62)
T(+)(P.V.<0.62)
T(-)

**E**

tSNE2

tSNE1

T(+) ●
T(-) ●

**F**

tSNE2

tSNE1

T(+)(P.V.>0.62) ●
T(+)(P.V.<0.62) ●
T(-) ●

**Figure 2.**

**Figure 2.  CRISPR-Cas9 gene perturbation profiling and identification of phenotypically perturbed cells.**

A    Schematic representation of the profiling of CRISPR-Cas9 gene perturbation phenotypes. Transient transfection of a targeting plasmid results in a mixed population of wild-type and genetically perturbed cells. Technical well-to-well variability can be accounted for by standardizing single-cell features to the wild-type cell population in every well. Logistic regression classifiers are fitted to the cell population to attempt to distinguish between T(+) and T(−) cells based on a set of single-cell features.

B, C    The predicted value (PV) is calculated for every cell in a well that was transiently transfected with a control targeting plasmid, or a *NUP160* targeting plasmid. A positive PV indicates classification into the phenotypically perturbed class. The dotted line indicates the threshold for further single-cell characterization [PV > 0.62 (mean + 3 × standard deviation of non-targeting control cells)].

D    Immunofluorescence image of mAb414 staining in HeLa cells transfected with a *NUP160* targeting plasmid. Cell outlines are coloured orange for T(+) cells that show a gene perturbation phenotype (PV > 0.62), red for T(+) cells with a PV < 0.62, blue for T(−) cells. Missegmented cells are outlined grey. Scale bar, 50 μm.

E, F    tSNE projection of cells transfected with a *NUP160* targeting plasmid. Single cells are colour coded according to tdTomato expression (E) and PV (F).

staining pattern, with the majority of signal localized to the nuclear periphery (Davis & Blobel, 1986). Within the T(+) population, we observe cells in which the mAb414 signal is mislocalized into a few bright foci, but we also find T(+) cells with wild-type mAb414 staining pattern. Importantly, a high PV distinguishes between the cells with wild-type and mislocalized mAb414 staining (Fig 2D). We further demonstrate this by plotting the cells into a two-dimensional projection of high-dimensional feature space using t-distributed stochastic neighbour embedding (Van Der Maaten & Hinton, 2008) (tSNE) (Fig 2E and F). T(−) cells localize to one region in multidimensional feature space, while T(+) cells are enriched in a different region, indicating that this region contains the phenotypically perturbed cells. Cells with a high PV exclusively localize to this region while a considerable fraction of T(+) cells localize to the region dominated by T(−) cells, indicating that these cells are phenotypically wild-type and should be ignored when characterizing the gene perturbation phenotype.

This approach now enables the profiling of genes involved in specific cellular processes by training classifiers based on specific sets of cellular features. To illustrate this, we first trained classifiers based on 86 features of cellular morphology and intensity and texture of the total protein stain (Table EV2). We chose a conservative threshold to select classifiers that score better than classifiers trained on non-targeting control populations and identified 49 perturbations including 14 perturbations that target proteasome subunits (Figs 3A and EV4A, Table EV4). We calculated the mean feature values of the phenotypically perturbed cells per well and discovered that the perturbation of proteasome subunits changes a broad set of cellular features (Fig EV4C). Next, we trained classifiers using an entirely different set of single-cell features, namely 118 features of the mAb414 staining pattern, and identified nine perturbations that target structural subunits of the NPC (Figs 3B and EV4B, Tables EV3 and EV5). These results indicate that we can

profile different dimensions of the multivariate cellular feature space by selecting different sets of single-cell features to identify genes that affect distinct biological processes. In addition, we analysed our screen by well-averaging the single-cell features to obtain mean feature profiles of T(+) and T(−) cells from each well in the experiment. We subsequently calculated the Mahalanobis distance between each profile and the total distribution of feature profiles to quantify phenotypic dissimilarity (Caicedo *et al*, 2017). Most of the hits identified in the between-well analysis overlap with the hits identified in the within-well analysis (Fig EV5). However, the within-well analysis identified more subunits of the proteasome complex when we profiled the cell morphology and total protein staining and more subunits of the NPC when we profiled the mAb414 staining features. This supports the notion that within-well profiling, by training computational classifiers to distinguish transfected from non-transfected cells, is more sensitive to detect phenotypic changes than a between-well comparison of well-averaged feature profiles.

To validate our results and further explore the power of image-based profiling of CRISPR-Cas9 gene perturbations in single cells, we focused on the NPC profiling. We constructed independent targeting plasmids for selected structural components of the NPC and *HSPA5/ Bip*, an ER chaperone involved in luminal ER protein folding and the regulation of the unfolded protein response (UPR; Pfaffenbach & Lee, 2011) that we identified in the profiling of both the mAb414 staining features as well as the cell morphology features (Table EV6). We transfected these constructs into HeLa cells, extracted single-cell features (Table EV7) and trained classifiers to separate T(+) from T(−) cells. To further characterize the gene perturbation phenotypes, we calculated mean feature profiles of the cells with high PV. Notably, by focussing our analysis specifically on the phenotypically perturbed cells, we obtain feature profiles in which phenotype-relevant features are more pronounced without reducing correlations

**Figure 3.  Large-scale image-based CRISPR-Cas9 gene perturbation profiling.**

A    Image-based profiling of the arrayed CRISPR-Cas9 library for perturbations affecting cellular morphology and total protein staining features. The classification score is a linear transformation of the misclassification error of logistic regression models trained to classify T(+) and T(−) cells. Perturbations targeting proteasome subunits or structural components of the NPC are colour-coded purple and green. Non-targeting control perturbations are colour-coded brown. The dotted line indicates the threshold used to select perturbations that have a higher classification score than non-targeting controls (third quartile + 1.5 × interquartile range of the classification scores of non-targeting controls). The size of the perturbation nodes is scaled according to the phenotypic score, which reflects the KS statistic calculated between the PV distributions of non-targeting control plasmid transfected cells and the transfected cells of the respective perturbation (see Materials and Methods).

B    Image-based profiling of mAb414 staining pattern. Colour coding and threshold calculation as in (A).

C    Hierarchical clustering of the standardized mean feature profiles of control cells or phenotypically perturbed cells transfected with plasmids targeting *HSPA5* or selected structural components of the NPC.

D    Immunofluorescence images and schematic representation of the mAb414 staining pattern in control cells or phenotypically perturbed cells from the *NUP62, HSPA5, NUP133, NUP107, NUP160* or *NUP98* targeted populations. Scale bar, 10 μm.

    

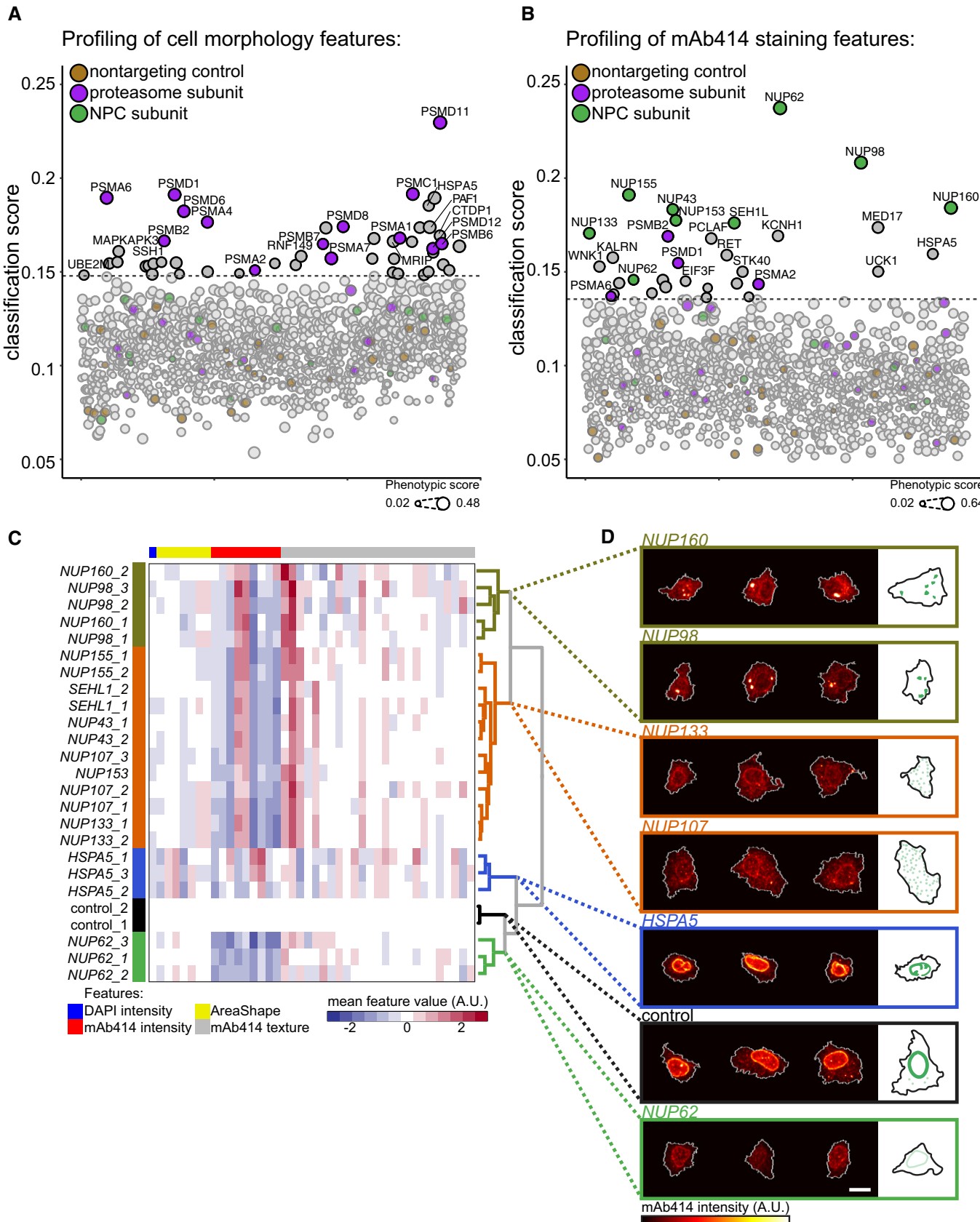

**Figure 3.**

between independent gRNAs targeting the same gene, indicating that the true gene perturbation phenotype is revealed (Fig EV6A and C). Strikingly, hierarchical clustering of these profiles, as well as the correlation between these profiles, revealed that profiles obtained from cells perturbed with different gRNAs targeting the same gene are highly similar across the full set of multivariate readouts (Figs 3C and EV6A and D), something that is generally not realized with RNAi (Collinet *et al*, 2010; Singh *et al*, 2015). The clustergram also demonstrates that different perturbations lead to different feature profiles. For instance, *NUP62*-targeted cells show a reduction in mAb414 staining intensity features (Figs 3C and D, and EV6A). This is expected as NUP62 is prominently bound by mAb414 (Davis & Blobel, 1986). *HSPA5*-targeted cells display a different phenotype. Here, the values of a broad set of features are altered, reflecting the smaller cell size, altered nuclear morphology and unusual staining pattern of mAb414 (Figs 3C and D, and EV6A). This  phenotype may reflect an early stage of the apoptotic programme, which could be triggered in the *HSPA5* knockout cells through ectopic activation of the UPR (Kihlmark *et al*, 2001; Pfaffenbach & Lee, 2011). The clustergram revealed that the perturbation profiles of *NUP160* and *NUP98* form a distinct cluster compared to the profiles of other components of the NPC (Weberruss & Antonin, 2016), which is caused by smaller differences across multiple mAb414 staining texture features (Figs 3C and D, and EV6A). Such a distinction is impossible to detect without multivariate profiling of single cells and is qualitatively confirmed by examining images of phenotypically perturbed cells. We observe a few bright foci of mAb414 staining in *NUP160*- and *NUP98*-knockout cells (Fig 3D), suggesting that central plug FG-NUPs coalesce into large aggregates in these cells. In contrast, in *NUP133*- or *NUP107*-knockout cells, the mAb414 signal localizes to small cytoplasmic foci (Fig 3D). This may reflect a relocalization of FG-NUPs to cytoplasmic membranous compartments termed annulate lamellae, as was previously observed in cells depleted of *NUP133* by RNAi (Walther *et al*, 2003).

In summary, we have combined large-scale CRISPR-Cas9 gene perturbation in multi-well plates, using transient transfection of targeting plasmids without any selection, with multivariate profiling of gene perturbation phenotypes in millions of single cells across thousands of genetic perturbations by means of automated microscopy and computer vision. By training classifiers that take into account the full cellular heterogeneity of specific subsets of cellular features, we identify genes involved in distinct cellular processes. We also developed a cost-effective pipeline to generate large-scale, arrayed libraries of sequence-verified CRISPR-Cas9 targeting plasmids that are available to the community. Because we analyse both perturbed and non-perturbed cells from the same well, our approach may also be applied to identify genes that have non-cell autonomous gene perturbation effects. Such genes could be identified by comparing wild-type cells from different wells, or training classifiers to distinguish wild-type cells that have genetically perturbed neighbouring cells with wild-type cells that are surrounded by wild-type neighbours. Although false-negative results are a general concern in high-throughput gene perturbation screens that only identify a perturbation if a phenotypic effect is observed, we identified several genetic perturbations that cause phenotypic changes in cellular morphology or the staining pattern of a marker of the NPC, indicating that our approach is a useful phenotypic screening tool. In the future, this may be addressed by combining image-based

phenotypic screening with smFISH, which provides an independent readout of whether the gene is perturbed. Furthermore, our approach facilitates the identification of phenotypically perturbed single cells for further analysis, which addresses the important issue that CRISPR-Cas9 does not functionally perturb every targeted cell. We show that image-based multivariate profiles of cells perturbed with independent gRNAs targeting the same gene are highly similar and we discovered distinct phenotypic effects when we profiled the staining pattern of a marker of the NPC. This work provides a framework for genome-scale multivariate profiling of microscopically resolved CRISPR-Cas9 induced gene perturbation phenotypes in mammalian cells.

# Materials and Methods

### Cell culture

HeLa cells were propagated from a single clone from the Kyoto strain, which was provided by J. Ellenberg (EMBL, Heidelberg). U2OS cells were obtained from the ATCC. Cells were cultivated in DMEM supplemented with 10% foetal bovine serum (FBS) (Gibco) at 37°C, 5% $CO_2$. Cells were tested for mycoplasma contamination. For the large-scale screen, cells in 384-well plates were cultivated in a Liconics rotating incubator to minimize plate positional effects.

### Plasmids

pSpCas9(BB)-2A-GFP (PX458) was a gift from Feng Zhang (Addgene plasmid #48138). To construct pSpCas9-2A-tdTomato-PAC (pRG84), 2A-tdTomato was PCR amplified from Addgene #54642 using primers  ggatccggagagggcagaggaagtctgctaacatgcggtgacgtcgaggagaatc ctggcccaatggtgagcaagggcgag and ggatcccttgtacagctcgtccatgc, subcloned into pJet and sequence verified. 2A-tdTomato was cloned into BamHI-digested lentiCRISPRv2 (Addgene plasmid #52961). Individual CRISPR-Cas9 targeting plasmids were constructed as described (Ran *et al*, 2013). Briefly, a pair of oligonucleotides was designed by prepending caccg to the 20-base pair gRNA sequence and prepending aaac and appending g to the reverse complement of the 20-base pair gRNA sequence. The oligos were annealed (5′ at 95°C, ramp down to 25°C at 2°C/min) and ligated into the BsmBI-digested pRG84 vector. All constructs were sequence verified by Sanger sequencing.

### Antibodies

Antibodies used in this study are as follows: mouse anti-CD71/ TFRC (BD Biosciences 555534), mouse anti-CD107a/LAMP1 (BD Biosciences 555798), mouse anti-YAP1 (Santa-Cruz 63.7), mouse anti-NPC (mAb414, Abcam), goat anti-mouse Alexa 488 highly cross-absorbed secondary antibody (Life Technologies A11029).

### CRISPR guide RNA sequence selection

We selected CRISPR guides using the Ensembl version GRCh38.78 gene annotation and the corresponding genome build. We avoided regions corresponding to either the first or the last exon in more than 25% of the annotated transcripts and selected guides with

Doench score at least 0.7 from different exonic regions of each gene. When sufficient candidate guides meeting these criteria were available, we chose guides shared by the maximal number of transcripts. Otherwise, we chose the guides with the best Doench score. The Doench score was calculated using the python script provided by Doench *et al* (2014). The script for selecting gRNA sequences is available as a Code EV1.

### Large-scale CRISPR-Cas9 screening library construction

Human genes associated with ubiquitination (gene ontology terms GO:0016567, GO:1990381, GO:0004843, GO:0031396, GO:1900044, GO:0016925) or phosphorylation (gene ontology terms GO:0016301, GO:0016791) were retrieved from Biomart. gRNA sequences were selected as described in the "CRISPR guide RNA sequence selection" paragraph. Oligos were designed by prepending the sequence GGAAAGGACGAAACACCG to the 20-base pair guide sequence and appending the sequence GTTTTAGAGCTAGAAATAGCAAGTTAA AATAAGGC. Array synthesized oligos were ordered from Custo-mArray (Bothell, WA, USA). The oligos were PCR amplified using Phusion polymerase (Thermo Scientific) with the primers TAACTTG AAAGTATTTCGATTTCTTGGCTTTATATATCTTGTGGAAAGGACG AAACACCG and ACTTTTTCAAGTTGATAACGGACTAGCCTTATTT TAACTTGCTATTTCTAGCTCTAAAAC. The PCR product was gel isolated and a Gibson assembly reaction with BsmBI-digested pRG84 was performed following the manufacturer's protocol (NEB). The reaction product was transformed into chemically competent Stbl3 cells (NEB) by heat shock. After 45 min recovery at 37°C in LB, the cells were plated on ampicillin-containing agar plates. The following day, individual colonies were transferred to 50 μl LB-amp in 96-well plates using sterilized toothpicks. Cultures were incubated overnight at 37°C on a shaking platform. We performed PCR-barcoding reactions for 71 plates of bacterial colonies. For every plate, each row of the plate contains one of eight forward primers (RG109, 110, 115–120) and one of 12 reverse primers (RG 111, 112, 121–130) (Table EV8). The PCR mix contained 0.25 μM forward and reverse primer, 0.25 μM dNTP (Sigma), 0.375 units of Taq polymerase (Sigma) in 15 μl 1× buffer (Sigma). Master mixes were prepared and dispensed into 96-well PCR plates using a Beckman Biomek FX liquid handling robot. PCR samples were transferred into the PCR mix using a 96-pin replicator. The replicator was sterilized by flaming with 96% ethanol between inoculations. 50 μl of 50% glycerol was added to the remainder of the culture before storing the cultures at −20°C. The PCR products were pooled per plate and gel isolated. A second barcode was introduced by PCR using one of the primers TSD501-TSD508 and one of the primers TSD701-TSD712 (Table EV8). The secondary PCR products were gel isolated, and 50 ng of each of the 71 secondary PCR products was pooled and processed for Illumina sequencing. Reads that could be mapped to the designed gRNAs were assigned to wells based on the barcodes. Only wells for which at least 50 reads were identified and the most abundant read was identified more than five times more often than the second most abundant read were selected and re-arrayed into 96 deep-well blocks (0.8 ml LB ampicillin per well) using a Beckman Biomek FX liquid handling robot. The cultures were covered with a gas-permeable seal and incubated overnight in a shaking incubator (330 rpm). The following day, glycerol stocks were prepared from the 50 μl of the cultures and the rest of culture

was collected by centrifugation. Transfection-grade plasmid DNA was isolated using Magnesil plasmid isolation kits (Promega). Plasmid concentrations were measured using a Tecan Infinity plate reader. 5.5 μl miniprep sample was diluted in 50 μl $H_2O$ containing 2 μg/μl DAPI. Plasmids were diluted to 10 ng/μl in Optimem (Gibco) in 384 deep-well blocks using a Beckman Biomek FX liquid handling robot, excluding the outer two wells of the plates. 10 μl plasmid solution was transferred to 384 well clear bottom plates and stored at −20°C before use.

### Reverse transfection

GeneJuice (EMD Millipore) was dissolved in Optimem (Gibco) in a ratio 2 μl GeneJuice: 1 μg plasmid DNA. The transfection mix was vortexed and incubated for 5 min at RT. The transfection mix was added to the plasmid DNA solution and mixed by pipetting or shaking of the plate for 1 s at 800 rpm on a thermomixer. The DNA-transfection mix was incubated for 10 min before the addition of the cell suspension (825 cells in 50 μl per well of a 384-well plate, 2,400 cells in 100 μl per well of a 96-well plate).

### Immunofluorescence

Cells were fixed in 4% paraformaldehyde (PFA, Electron Microscopy Sciences) for 20 min at room temperature (RT). Cells were permeabilized for 15 min in 0.2% Triton X-100 and blocked in 5% goat serum (Cell Signaling Technology). If S-phase labelling was performed, cells were incubated for 15 min with 200 μM Edu in culture medium prior to fixation and a Click-iT Edu Alexa-647 (Thermo Scientific) labelling reaction was performed according to manufacturer's instructions before incubation with a primary antibody in 5% goat serum for 1 h at RT. Cells were washed 3× with phosphate-buffered saline (PBS) and incubated with secondary antibody for 1 h followed by 3 PBS washes. DNA was stained using DAPI (0.1 μg/ml in PBS) for 10 min. Total protein was stained with succinimidyl-ester-Alexa-647 for 5 min [1:200,000 in carbonate buffer (0.1 M $NaHCO_3$, 25 mM $Na_2CO_3$)].

### Single-molecule mRNA FISH

Branched DNA FISH was performed as described in Battich *et al* (2013). Gene-specific probe pairs were obtained from Affymetrix.

### Image acquisition and single-cell feature quantification

Images were acquired on a Yokogawa CellVoyager 7000 automated microscope equipped with a CSU-X1 spinning disc, Neo sCMOS cameras (Andor) and UPLSAPO 20× (NA 0.75, Olympus) lens. CellProfiler software was used for image analysis, cell segmentation and single-cell feature quantification as described in Stoeger *et al* (2015). We segmented the nuclear periphery by expanding and shrinking the nucleus segmentation by 5 pixels. We segmented the cytoplasm by masking the cell segmentation by the expanded nucleus. The CellProfiler pipeline is available as Dataset EV2. We employed CellClassifier (https://www.pelkmanslab.org/?page_id= 63) for data clean up and classification of transfected cells and cells in S-phase of the cell cycle. We excluded missegmented cells, mitotic cells and cells displaying staining artefacts from further

analysis (Stoeger *et al*, 2015). Computations were performed on the Brutus computing cluster (ETH Zürich) using the task manager iBRAIN.

### Phenotypic profiling by between-well comparison of feature profiles

Mean feature profiles were obtained for the T(+) and T(−) cell populations per well (the features used are listed in Tables EV2 and EV3). Feature profiles were standardized by median B-score to correct for plate positional effects (Caicedo *et al*, 2017). The Mahalabobis distance between each feature profile and the distribution of all profiles was calculated and used as a measure for phenotypic dissimilarity.

### Phenotypic profiling by within-well classification of transfected and non-transfected cells

Single-cell features of the mAb414 staining pattern (intensity and texture features in cells, nuclei, cytoplasm and nuclear periphery) or features of the area and shape of the cells and nuclei and intensity and texture features of the total protein stain were standardized by the mean and standard deviation of the T(−) cell population per well. We excluded wells with fewer than 300 transfected cells from further analysis. As a first step in the screen analysis, the dimensionality of the data set was reduced by principal component analysis. The features used for the PCA are listed in Tables EV2 and EV3. We selected the first 50 (for the cell morphology profiling) or 30 (for the mAb414 staining features profiling) principal components of the data sets. We randomly selected 500 T(+) and T(−) cells (with replacement) form every targeted cell population and trained a 10-fold cross-validated logistic regression model on the single-cell data using the R software package glmnet (Friedman *et al*, 2010). We employed the least absolute shrinkage and selection operator (LASSO) method for feature selection and bootstrapped this procedure 100 times. We averaged the misclassification error per perturbation. The classification score is a linear transformation of the average misclassification error of the models obtained in the bootstraps (we multiply the mean misclassification error with −1 and add 0.5). We chose the third quantile + 1.5 × the interquartile range of the classification score of models trained on non-targeting control transfected populations as a conservative threshold to select classifiers that perform better than classifiers trained on non-targeting control perturbations. For every logistic regression model trained, the PV was calculated for every cell in the well. We averaged the PV per cell over all bootstraps. To calculate the phenotypic score for each perturbation, we calculated the Kolmogorov–Smirnov statistic between the distribution of PV of transfected cells from non-targeting control plasmid transfected wells and the distribution of PV of transfected cells from the respective targeted population.

We calculated the enrichment of GO terms associated with the top-scoring perturbations relative to the GO terms associated with the genes that were represented in the arrayed CRISPR-Cas9 library and calculated *P*-values using a hypergeometric test.

In the validation experiments of selected hits from the mAb414 profiling screen, we analysed the features listed in Table EV7. We reduced the dimensionality of the mAb414 staining texture features

by principal component analysis prior to calculating the mean feature values of all phenotypically perturbed cells per perturbation. We standardized the mean feature profiles to the mean feature values of control cells.

### Data and software availability

The CellProfiler pipeline is available as Dataset EV2. The script used for selecting gRNA sequences is provided as Code EV1.

**Expanded View** for this article is available online.

## Acknowledgements
We thank Lucy Poveda and Weihong Qi of the FGCZ for the sequencing and bioinformatics analysis of the PCR barcoding of the arrayed screening library. RdG was supported by a SystemsX.ch TFdP fellowship (SystemsX.ch 2015/342). We thank the members of the Pelkmans laboratory for critically reading the manuscript.

## Author contributions
LP and RG conceived and designed the project. RG and JL carried out experiments and analysed data. HL developed the gRNA selection script. RH assisted in the construction of the arrayed CRISPR screening library. RG and LP wrote the manuscript.

## Conflict of interest
The authors declare that they have no conflict of interest.

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
