## [Review Process File · Molecular Systems Biology]

Large-scale image-based profiling of single-cell phenotypes in arrayed CRISPR-Cas9 gene perturbation screens

Reinoud de Groot, Joel Lüthi, Helen Lindsay, René Holtackers & Lucas Pelkmans

Review timeline:

Submission date:	23 October 2017
Editorial Decision:	27 November 2017
Revision received:	18 December 2017
Accepted:	21 December 2017

Editor: Maria Polychronidou

Transaction Report:

1st Editorial Decision

27 November 2017

Thank you again for submitting your work to Molecular Systems Biology. We have now heard back from the two referees who agreed to evaluate your study. As you will see below, the reviewers are overall quite positive and think that the proposed methodology seems interesting. They raise however a series of concerns, which we would ask you to address in a revision of the manuscript.

The reviewers' recommendations are rather clear and therefore I think that there is no need to repeat all the points listed below. Please let me know in case you would like to discuss further any of the issues raised by the reviewers.

REVIEWER REPORTS

Reviewer #1:

de Groot et al. describe a method for making a large microscopy based screen of genetic perturbations in human cells.

I strongly believe that library scale microscopy based phenotypic screens will be very important for systems biology. The challenges of making all the steps in these assays working sufficiently well are exceptional, which is why this study should be considered for publication in MSB despite the limitations described below.

The study is conceptually similar to Shi H et al. and Kuwada NJ et al. although the mammalian aspect of the study adds a valuable dimension and the problem at it is unclear which cells are actually perturbed. The differences and similarity to these references should be discussed.

The approach to keep all strains isolated in different wells makes the assay very laborious (and expensive) including pipetting robots and many plates in several steps. The authors should consider the possible advantages of making the whole screen pooled by genotyping the library in situ after phenotyping as was recently described by Lawson et al and Emanuel et al.

My major concern with the study is that it, as far as I understand, is not possible to know if a specific perturbation has worked unless a significant phenotype is observed. That is, the method can be used as a screening tool for identifying genes that gives significant changes in phenotypes that can be observed by microscopy, but it can not be used to determine which is the phenotype of a specific perturbation. If the authors agree, I believe this should be spelt out clearly.

It is also important to avoid circular arguments based on this limitation. For example Line 119 " We characterized the mean feature values of the phenotypically perturbed cells and discovered that the perturbation of proteasome subunits changes multiple cellular morphology and total protein staining features" Isn't it obvious that cells that are characterized as being perturbed based on some morphology features display morphology phenotypes?

Minor concerns

In the sentence " 72% of the targeting plasmids achieved a single-cell knockout efficiency of >30%, indicating that we can reliably select gRNA sequences that perturb gene expression" What does knockout efficiency mean and how is 30% selected ?

Script for selecting gRNA sequences should be uploaded.

Refs

Shi H, Colavin A, Lee TK, Huang KC (2017) Strain library imaging protocol for high-throughput, automated single-cell microscopy of large bacterial collections arrayed on multiwell plates. *Nat Protoc* 12: 429 - 438

Kuwada NJ, Traxler B, Wiggins PA (2015) Genome-scale quantitative characterization of bacterial protein localization dynamics throughout the cell cycle. *Mol Microbiol* 95: 64 - 79

Emanuel G, Moffitt JR Zhuang X High-throughput, image-based screening of pooled genetic-variant libraries. *Nat Methods*. 2017

Lawson MJ, Camsund D, Larsson J, Baltekin Ö, Fange D, Elf J. In situ genotyping of a pooled strain library after characterizing complex phenotypes. *Mol Syst Biol*. 2017

Reviewer #2:

Summary

The work titled "Large-scale image-based profiling of single-cell phenotypes in 2 arrayed CRISPR-Cas9 gene perturbation screens" by Reinoud de Groot and colleges describes an approach to using CRISPR gene editing techniques in image based high throughput experiments.

The authors developed a cost-effective method to construct an arrayed library for CRISPR-Cas9 perturbation (1,423 genes; 2,281 plasmids) and worked out methods to treat cells with these reagents in image-based format.

They use a target plasmid that encodes a fluorescent protein (tdTomato) to distinguish between transfected cells (T+) and all non transfected cells (T-). Using automated microscopy, all non transfected cells are identified and used as a control population within each well. Next, the effect of a perturbation is quantified using fluorescence microscopy. Automated image analysis is used to extract a morphological profile of the specific staining for each single cell. Given these profiles and the information whether a cell is transfected or not, a classifier is trained to separate all perturbed and non transfected cells. The accuracy of this classifier is used to implement a measure for the strength / effect of the perturbation. For all perturbations with a high classification score, a well-based morphological profile is created that can be used in the downstream analysis. Furthermore, the analyzed experiments reveal that the method is sensitive enough to identify and distinguish the effect of the knockout of different genes related to NPC and allows to quantify these effects using microscopical image data.

Overall, the presented method allows CRISPR based perturbation to be used in image based profiling experiments and other readouts that require an arrayed format. As compared to pooled RNAi based experiments that generally rely on cell viability as the readout, the presented method allows harvesting the rich information available in image based assays at a single cell level, including interpretable features (size of the cell, deformation of the nuclei,...)

Major points

As mentioned above, given that the authors are using a transfection control that is visible at the single cell level, they can employ a very useful trick: they can use untransfected cells within the well to control for batch effects/technical variation. This is in contrast to prior image based high throughput experiments which typically use control populations of non-transfected cells that are completely untreated (<https://www.nature.com/articles/nmeth.4397>) The method appears to be effective and presumably enhances the sensitivity of the assay. One note, however, is that it should be mentioned that transfected cells can influence non-transfected cells within the well and that such effects may skew results in some samples. The authors might be able to estimate the rate at which this occurs by comparing the morphological profile of a large number of non infected T(-) populations with non-treated control populations to see whether any consistent differences are seen across a large number of genes. It is unclear whether the present experiment has sufficient experimental samples to do such an analysis.

The authors state: "profiles obtained from cells perturbed with different gRNAs targeting the same gene are highly similar across the full set of multivariate readouts (Figure 3C,S5A), something that is generally not realized with RNAi (Collinet et al, 2010)." If I understand correctly, the conclusion seems to be a qualitative assessment based on only 9 genes. It is unclear why this analysis was not done quantitatively, that is, measuring the correlation between targeting-same-gene vs other-genes. As well an additional reference for this is Singh et al. (Morphological Profiles of RNAi-Induced Gene Knockdown Are Highly Reproducible but Dominated by Seed Effects).

The hit rate of 49 perturbations out of 1,400 that induce a detectable change in morphology seems very low compared to the 50% hit rate in the only comparable paper I'm aware of, which used gene overexpression (Rohban, eLife, 2017). The authors might comment on this (or any other genetic perturbation screens looking for general changes in morphology) to compare and contrast with their results.

Finally, the paper does not seem to present any dramatic biological discoveries, or at least it is not made clear in the text which of the genes found to be "hits" in the two screens are novel or interesting. Nevertheless, in my view the methodological advances should be quite useful and interesting to the scientific community.

Minor points

The morphological analysis uses PCA for dimension reduction as well as a feature selection and it is not always clear, whether principal components or features are used. The manuscript would benefit from a consistent naming.

L41 It is not clear the first paragraph of the Results section is describing controls, some context

would be helpful for understanding that this paragraph does not describe the screen itself.

L132-133. Please clarify if the mean values were calculated per well or if replicates could be used (if any existed).

L282. Please add the used version of CellClassifier. Additionally, it would be great if you could provide the implemented pipeline as a repository

L 293 - 298. Two methods for dimension reduction are used, PCA and feature selection. It is not clear, whether the principal components have been used as input for the feature selection or two feature selection have been performed (one feature selection based on the PCA, one based on the described LASSO method).

L 311 - 314. Please clarify which experiment is denoted with validation experiment and if the feature extraction differs from the method described in L 293-298.

L 457 + 477 You refer to "supplementary methods" which does not exist. Probably you refer to the "Methods" section. Please correct.

Figure 3c. Please explain which features were used to create this plot (i.e. did you use principle components of a PCA, selected features or all features)

Figure S22. The pie chart lists 1450 genes, 2188 plasmids. However, the text lists 2281 Plasmids and 1423 genes. Please correct.

Figure S4a. p-value for nuclear pore is missing.

Figure S4c. Please specify which features are shown (selected features, best features of the PCA, all features?)

Figure S5. The legend says "Boxplots of the standardized single cell features". However, principal components are listed (mAb414 Texture PC1). Please correct.

Table S2.

Why are some target genes are missing? Please comment on the entry "somethingWrong".

Table S3.

Each column has two descriptions (first column: 1. "Features morphology profiling:" and 2. "50 principle components of the features:"). Please clarify if all features used for the pca are listed or selected features of the resulting 50 principal components are listed.

The authors state "Our script for selecting gRNA sequences is available on request." If they are willing to distribute it, it should be made available as supplemental material, or run the risk of being unavailable to others' research. Studies show that requesting material "available upon request" has a very low rate of success.

The language in the title referring to 2 screens is confusing. It is not conventional to describe a single experiment measured for 2 phenotypes as 2 separate screens.

In one place, the number of cells per population is described as 10^4 and in two other places as 4×10^3 . Why not just state " $\sim 4,000$ " in each place, to be more precise and consistent?

Check spelling throughout "principle components"

We would like to thank the reviewers for a careful and constructive review. We have revised our manuscript to address all reviewer comments. We feel that, by incorporating the suggestions of the reviewers, we have significantly improved our manuscript. Below is a point-by-point response to the issues raised by the reviewers, *we have included edits in the revised version of the manuscript in italic.*

Reviewer #1:

de Groot et al. describe a method for making a large microscopy based screen of genetic perturbations in human cells.

I strongly believe that library scale microscopy based phenotypic screens will be very important for systems biology. The challenges of making all the steps in these assays working sufficiently well are exceptional, which is why this study should be considered for publication in MSB despite the limitations described below.

The study is conceptually similar to Shi H et al. and Kuwada NJ et al. although the mammalian aspect of the study adds a valuable dimension and the problem at it is unclear which cells are actually perturbed. The differences and similarity to these references should be discussed.

Shi et al. describe an experimental approach that facilitates imaging of bacterial cells cultured in multiwell plates by transferring them to an agar pad followed by automated microscopy. Kuwada et al. quantified fluorescent fusion protein localization in live bacterial cells, they grow the bacteria in multiwell plates and transfer them to an agarose pad prior to imaging. Similar to these approaches, we image many populations of cells in parallel. However, we use populations of genetically perturbed mammalian cells in our experiments. Our experimental strategy differs in another key aspect, in our approach, mammalian cells are seeded in multiwell plates with a clear plastic bottom. We image the cells through the bottom of the multiwell plates using an automated microscope. In this way, we can prepare libraries of reagents to perturb individual genes, arrayed in multiwell plates and we do not need to transfer the cells prior to imaging.

The approach to keep all strains isolated in different wells makes the assay very laborious (and expensive) including pipetting robots and many plates in several steps. The authors should consider the possible advantages of making the whole screen pooled by genotyping the library in situ after phenotyping as was recently described by Lawson et al and Emanuel et al.

The strategy that is developed by Lawson et al. and Emanuel et al. entails that gene perturbations are performed in a pooled format (which is experimentally much less demanding than our arrayed screening strategy), followed by in-situ genotyping of cells after a phenotypic profile is obtained. However, in-situ genotyping of cells is experimentally challenging whereas an arrayed screening strategy allows the direct identification of the targeted gene in every well by interrogating the annotation of the screening library. Moreover, our approach scales particularly well when it comes to quantifying the phenotypic response of the same perturbation in large numbers of single mammalian cells. We discuss the experimental strategy that was developed Lawson et al. and Emanuel et al. in the revised manuscript:

Recently, methods to perturb cells in a pooled format, followed by image-based phenotyping and in-situ genotyping were developed for prokaryotic model systems (Lawson et al, Emanuel et al.). An alternative screening strategy involves seeding cells in multi-well plates that contain reagents that perturb one specific gene per well. This arrayed screening strategy allows detailed, image-based phenotyping of populations of cells in which specific genes are perturbed.

My major concern with the study is that it, as far as I understand, is not possible to know if a specific perturbation has worked unless a significant phenotype is observed. That is, the method can be used as a screening tool for identifying genes that gives significant changes in phenotypes that

can be observed by microscopy, but it can not be used to determine which is the phenotype of a specific perturbation. If the authors agree, I believe this should be spelt out clearly.

We agree with the reviewer. The methods that we describe are indeed aimed to be used as a screening tool. False negative results are always a concern in functional genomic screens, but the fact that we identified genetic perturbations that cause several classes of phenotypic effects when we profiled a marker of the nuclear pore complex indicates that our approach can be used for screening purposes. Moreover, we here show that smFISH could in principle be used to address this issue, as it gives a phenotype-independent readout of whether a gene is successfully perturbed. We point this out more clearly in the revised manuscript:

Although false-negative results are a general concern in high-throughput gene perturbation screens that only identify a perturbation if a phenotypic effect is seen, we identified several genetic perturbations that cause phenotypic changes in cellular morphology or the staining pattern of a marker of the NPC, indicating that our approach is a useful phenotypic screening tool. In the future, this may be addressed by combining image-based phenotypic screening with smFISH, which provides an independent readout of whether the gene is perturbed.

It is also important to avoid circular arguments based on this limitation. For example Line 119 " We characterized the mean feature values of the phenotypically perturbed cells and discovered that the perturbation of proteasome subunits changes multiple cellular morphology and total protein staining features" Isn't it obvious that cells that are characterized as being perturbed based on some morphology features display morphology phenotypes?

We apologize for this confusion. The point that we tried to make here is that the genetic perturbation of proteasome subunits changes multiple cellular features (as opposed to the possibility that the targeting of proteasome subunits would only affect a single feature). We made the following change to the revised manuscript to point this out more clearly:

We characterized the mean feature values of the phenotypically perturbed cells and discovered that the perturbation of proteasome subunits changes a broad set of cellular features (Figure EV4C).

Minor concerns

In the sentence " 72% of the targeting plasmids achieved a single-cell knockout efficiency of >30%, indicating that we can reliably select gRNA sequences that perturb gene expression" What does knockout efficiency mean and how is 30% selected ?

With the term knockout efficiency, we refer to the percentage of transfected cells that have lost expression of the targeted gene, we have made this explicit in the revised version of the manuscript:

We targeted 26 genes with three targeting plasmids each. 72% of the targeting plasmids perturbed gene expression in more than 30% of transfected cells, indicating that we can reliably select functional gRNA sequences (Figure 1H, Table EV1).

The threshold to classify a targeting plasmid as functional was set at 30% because this fraction phenotypically perturbed cells would be result in a classification score of 0.15 when a logistic regression model would be fitted. This would be sufficient for a perturbation to be classified as a 'hit' given the threshold we selected in the large-scale profiling experiment.

Script for selecting gRNA sequences should be uploaded.

The gRNA selection script and instructions for its use are supplied as a supplementary file: deGroot_gRNAselection.zip.

Refs

Shi H, Colavin A, Lee TK, Huang KC (2017) Strain library imaging protocol for

high-throughput, automated single-cell microscopy of large bacterial collections arrayed on multiwell plates. *Nat Protoc* 12: 429 - 438

Kuwada NJ, Traxler B, Wiggins PA (2015) Genome-scale quantitative characterization of bacterial protein localization dynamics throughout the cell cycle. *Mol Microbiol* 95: 64 - 79

Emanuel G, Moffitt JR Zhuang X High-throughput, image-based screening of pooled genetic-variant libraries. *Nat Methods*. 2017

Lawson MJ, Camsund D, Larsson J, Baltekin Ö, Fange D, Elf J. In situ genotyping of a pooled strain library after characterizing complex phenotypes. *Mol Syst Biol*. 2017

Reviewer #2:

Summary

The work titled "Large-scale image-based profiling of single-cell phenotypes in 2 arrayed CRISPR-Cas9 gene perturbation screens" by Reinoud de Groot and colleges describes an approach to using CRISPR gene editing techniques in image based high throughput experiments.

The authors developed a cost-effective method to construct an arrayed library for CRISPR-Cas9 perturbation (1,423 genes; 2,281 plasmids) and worked out methods to treat cells with these reagents in image-based format.

They use a target plasmid that encodes a fluorescent protein (tdTomato) to distinguish between transfected cells (T+) and all non-transfected cells (T-). Using automated microscopy, all non-transfected cells are identified and used as a control population within each well. Next, the effect of a perturbation is quantified using fluorescence microscopy. Automated image analysis is used to extract a morphological profile of the specific staining for each single cell. Given these profiles and the information whether a cell is transfected or not, a classifier is trained to separate all perturbed and non transfected cells. The accuracy of this classifier is used to implement a measure for the strength / effect of the perturbation. For all perturbations with a high classification score, a well-based morphological profile is created that can be used in the downstream analysis. Furthermore, the analyzed experiments reveal that the method is sensitive enough to identify and distinguish the effect of the knockout of different genes related to NPC and allows to quantify these effects using microscopical image data.

Overall, the presented method allows CRISPR based perturbation to be used in image based profiling experiments and other readouts that require an arrayed format. As compared to pooled RNAi based experiments that generally rely on cell viability as the readout, the presented method allows harvesting the rich information available in image based assays at a single cell level, including interpretable features (size of the cell, deformation of the nuclei,...)

Major points

As mentioned above, given that the authors are using a transfection control that is visible at the single cell level, they can employ a very useful trick: they can use untransfected cells within the well to control for batch effects/technical variation. This is in contrast to prior image based high throughput experiments which typically use control populations of non-transfected cells that are completely untreated (<https://www.nature.com/articles/nmeth.4397>) The method appears to be effective and presumably enhances the sensitivity of the assay. One note, however, is that it should be mentioned that transfected cells can influence non-transfected cells within the well and that such effects may skew results in some samples. The authors might be able to estimate the rate at which this occurs by comparing the morphological profile of a large number of non infected T(-) populations with non-treated control populations to see whether any consistent differences are seen

across a large number of genes. It is unclear whether the present experiment has sufficient experimental samples to do such an analysis.

The reviewer points out that the knockout of specific genes in perturbed cells may cause phenotypic changes in unperturbed wild-type cells in the same well. Genes encoding secreted signalling molecules, cell-adhesion proteins or extracellular matrix components might be expected to have such non-cell autonomous effects. Our experimental approach would indeed allow the identification of such genes. One possibility, as suggested by the reviewer, is to compare non-transfected cells (which are genetically wild-type) from different wells. To address this point, we have calculated the mean feature profile of transfected, as well as non-transfected cells for every well of our screen for mAb414 staining features and the cell shape and total protein staining features. We corrected the feature profiles for plate positional effects by B-scoring and calculated the Mahalanobis distance of each feature profile from the distribution of all feature profiles to quantify phenotypic dissimilarity (Figure R1A,C). The top scoring profiles from this analysis agree with the top scoring perturbations identified based on the classification score as identified in the within-well analysis (Figure R1B,D). Notably, HSPA5 and SEH1L were not identified as a hit based on the Mahalanobis distance calculated based on the mAb414 staining features, but were identified based on the classification score and were confirmed in our follow up experiments. This is consistent with the notion of the reviewer that a within-well comparison can indeed be more sensitive to phenotypic changes than a between-well comparison. In our dataset, we did not observe profiles from non-transfected cells to be significantly different from the total distribution of phenotypic profiles (there are no grey dots in Fig R1A,C above the threshold), indicating that there are no clear examples of gene perturbations that cause non-cell autonomous effects in the current screen. This is most likely explained by the phenotypic assays that we profiled. We hypothesize that profiling of components of intercellular signalling networks is better suited to identify genes that act non-cell autonomously.

Figures R1: Phenotypic profiling by between-well comparison of feature profiles.

A,C: The mean mAb414 feature profiles (A) or cell morphology features (C) were calculated for T(+) and T(-) cells per well. For each profile, the Mahalanobis distance from the distribution of all profiles was calculated. Nodes represent feature profiles, color-coded magenta and grey for profiles obtained from T(+) and T(-) cells respectively. The dotted line indicates the threshold used to select perturbations that have large distance from non-targeting controls (third quartile + 3*interquartile range of the distance of non-targeting controls). Nodes are scaled according to the classification score which is based on the within-well comparison of T(+) and T(-).

B,D: The Mahalanobis distance of T(+) profiles from the total distribution of mean feature profiles was plotted against the classification score (as obtained from within-well comparison of T(+) and T(-) cells) for the profiling of the mAb414 features (B) and cell morphology features (D).

We have added a few lines to the revised manuscript to make these points and we added figure R1 to our manuscript as supplementary figure EV5:

In addition, we analysed our screen by well-averaging the single cell features to obtain mean feature profiles of T(+) and T(-) cells from each well in the experiment. We subsequently calculated the Mahalanobis distance between each profile and the total distribution of feature profiles to quantify phenotypic dissimilarity (Caicedo et al, 2017). Most of the hits identified in the between-well analysis overlap with the hits identified in the within-well analysis (Figure EV5). However, the within-well analysis identified more subunits of the proteasome complex when we profiled the cell morphology and total protein staining and more subunits of the NPC when we profiled the mAb414 staining features. This supports the notion that within-well profiling, by training computational classifiers to distinguish transfected from non-transfected cells, is more sensitive to detect phenotypic changes than a between-well comparison of well-averaged feature profiles.

An alternative approach to identify genes that act non-cell autonomously in neighbouring cells uses within-well comparisons. Here, we train classifiers to distinguish cells with a transfected neighbouring cell from cells that do not have transfected neighbours. We performed this analysis, but we did not identify genes that cause significant non-cell autonomous gene perturbation effects in neighbouring cells in our phenotypic assays. We have added a few lines to the discussion of our manuscript to point out the possibility of screening for non-cell autonomous gene perturbation effects:

Because we analyse both perturbed and non-perturbed cells from the same well, our approach may also be applied to identify genes that have non-cell autonomous gene perturbation effects. Such genes could be identified by comparing wild-type cells from different wells, or training classifiers to distinguish wild-type cells that have genetically perturbed neighbouring cells with wild-type cells that are surrounded by wild-type neighbours.

The authors state: "profiles obtained from cells perturbed with different gRNAs targeting the same gene are highly similar across the full set of multivariate readouts (Figure 3C,S5A), something that is generally not realized with RNAi (Collinet et al, 2010)." If I understand correctly, the conclusion seems to be a qualitative assessment based on only 9 genes. It is unclear why this analysis was not done quantitatively, that is, measuring the correlation between targeting-same-gene vs other-genes. As well an additional reference for this is Singh et al. (Morphological Profiles of RNAi-Induced Gene Knockdown Are Highly Reproducible but Dominated by Seed Effects).

To address this point, we calculated the Pearson's correlation coefficient between mean feature profiles obtained from cells perturbed with gRNAs targeting the same gene, or different genes. We first analysed this for the follow-up experiment where we systematically used multiple gRNAs against the same gene for multiple genes. We find very high correlations between profiles when the gRNAs target the same gene, and these are higher than when the gRNAs target different genes. That the latter correlations were not zero was furthermore expected, since the genes that we selected for follow-up (with the exception of 1) encode for proteins that are part of the same macromolecular complex (the nuclear pore). When this correlation analysis was performed using multivariate profiles obtained with gRNAs targeting different genes from the primary screen, and which are thus not biased towards functionally related genes, the correlations indeed centre around zero. This nicely shows that multivariate phenotypic profiles obtained with CRISPR-Cas9-mediated gene perturbation are highly consistent amongst multiple gRNAs targeting the same gene, and that this consistency is so high that it allows capturing differences between the phenotypic effects of knocking out subunits of the same macromolecular complex. We have included Figure R2 as supplementary Figure EV6D in the manuscript. We have also included the suggested reference.

Figure R2: Boxplots of Pearson's correlation coefficients calculated between mean feature profiles of phenotypically perturbed cells transfected with plasmids targeting the same gene, or different genes. Phenotypic profiles were obtained from cells transfected with plasmids targeting selected subunits of the NPC and HSPA5 (green) or the top-scoring genes that were identified in the large-scale profiling of the mAb414 staining features (grey).

The hit rate of 49 perturbations out of 1,400 that induce a detectable change in morphology seems very low compared to the 50% hit rate in the only comparable paper I'm aware of, which used gene overexpression (Rohban, eLife, 2017). The authors might comment on this (or any other genetic perturbation screens looking for general changes in morphology) to compare and contrast with their results.

The key difference between the experimental approach of Rohban et al. and our experiments is that Rohban et al. used features obtained with the CellPainting assay to characterize cell morphology. CellPainting allows the visualisation of the cell and nucleus morphology, but additionally visualizes the endoplasmic reticulum, nucleolus, cytoplasmic RNA, the actin cytoskeleton, Golgi complex and mitochondria. In contrast, we only profiled cell shape and the total protein stain in our experiments. Rohban et al. can therefore evaluate many more aspects of cell physiology than we did in our approach. Secondly, the genes that Rohban et al. overexpressed were selected based on the fact that mutated alleles of these genes are found in human cancers. For these reasons, it is perhaps not surprising that Rohban et al. found that a higher fraction of genes in their library change cellular features when overexpressed, compared to our observations in loss-of-function experiments.

The lab of Chris Bakal has addressed the genetic control of cell morphology. In Yin et al. (2013) an RNAi screen for the morphology of Drosophila haemocytes is described. Here, 191 out of the 280 kinases and phosphatases screened change the distribution of cell shapes in the targeted cell population. We hypothesize the main reason for the contrast between our results and the observations of Yin et al. lies in the experimental model system. We use HeLa cells cultured on a plastic surface in our experimental setup. We observe very little variability in cell shape in HeLa cells in these conditions. It has been hypothesized that cells of epithelial origin, such as HeLa cells, are particularly robust to changes in cell shape (Olson, 2013). A screen focussed on identifying genes that control cell shape would benefit from using a cell model that displays variability in cell shape in standard culture conditions (such as fibroblasts). In addition, growing cells on a specific extracellular matrix or a softer substrate might allow more variability in cell shapes and therefore improve the possibilities to detect genetic perturbations that control cell shape. We decided not to further comment on this in the main text of the revised manuscript as most of the follow-up was dedicated to the hits found in the screen using nuclear pore complex staining.

Yin Z, Sadok A, Sailem H, McCarthy A, Xia, X, Li F, Garcia MA, Evans L, BarrAR, Perrimon N, Marshall CJ, Wong ST, Bakal C, A screen for morphological complexity reveals regulators of switch-like transitions between discrete cell shapes. Nat Cell Biol. 2013, 15(7):860-71

Olson MF, Finding the shape-shifter genes. Nat Cell Biol. 2013, 15(7):723-5

Finally, the paper does not seem to present any dramatic biological discoveries, or at least it is not made clear in the text which of the genes found to be "hits" in the two screens are novel or interesting. Nevertheless, in my view the methodological advances should be quite useful and interesting to the scientific community.

Minor points

The morphological analysis uses PCA for dimension reduction as well as a feature selection and it is not always clear, whether principal components or features are used. The manuscript would benefit from a consistent naming.

L41 It is not clear the first paragraph of the Results section is describing controls, some context would be helpful for understanding that this paragraph does not describe the screen itself.

We changed the revised manuscript to better emphasize that we describe separate experiments here.

We devised an experimental strategy for the application of the CRISPR-Cas9 system in an arrayed screening format. To allow maximum flexibility with regards to the cell line and assay used for screening, we opted for a one-component system where the coding sequence for SpCas9, a chimeric gRNA and a fluorescent protein (tdTomato) are combined on a single plasmid. We introduced targeting plasmids into human tissue culture cells by reverse transfection and assayed expression of the targeted gene by quantitative immunofluorescence (Figure 1A). As a proof of concept, we targeted the Transferrin Receptor (TFRC) in HeLa cells and assessed TFRC expression in approximately 4000 single cells per experimental condition.

L132-133. Please clarify if the mean values were calculated per well or if replicates could be used (if any existed).

The mean feature profiles of the phenotypically perturbed cells (i.e. the cells with a high PV) were calculated per well. We have clarified this point in the revised manuscript:

We characterized the mean feature values of the phenotypically perturbed cells and discovered that the perturbation of proteasome subunits changes a broad set of cellular features (Figure EV4C).

L282. Please add the used version of CellClassifier. Additionally, it would be great if you could provide the implemented pipeline as a repository

The code for CellClassifier and the custom CellProfiler modules are available on GitHub <https://github.com/pelkmanslab/>. We have added this information to the **Image acquisition and single cell feature quantification section of the Material and Methods and the CellProfiler pipeline is supplied as a supplementary file deGroot_CPPpipeline.txt.**

L 293 - 298. Two methods for dimension reduction are used, PCA and feature selection. It is not clear, whether the principal components have been used as input for the feature selection or two feature selection have been performed (one feature selection based on the PCA, one based on the described LASSO method).

We performed PCA to reduce the number of dimensions before we fit logistic regression models. We employed the LASSO method to reduce variance in the fitted models, we did not further exploit the fact that the LASSO method can be used for feature selection.

L 311 - 314. Please clarify which experiment is denoted with validation experiment and if the feature extraction differs from the method described in L 293-298.

In the validation experiments we further characterized the gene perturbation phenotype of selected hits of the mAb414 feature profiling screen. We analysed the single cell features listed in figure EV6 in this experiment. For clarity, we have listed these features in table EV7 and point this out in the revised manuscript:

In the validation experiments of selected hits from the mAb141 profiling screen, we analysed the features listed in Table EV7. We reduced the dimensionality of the mAb414 staining texture

features by principal component analysis prior to calculating the mean feature values of all phenotypically perturbed cells per perturbation.

L 457 + 477 You refer to "supplementary methods" which does not exist. Probably you refer to the "Methods" section. Please correct.

Apologies, we corrected this.

Figure 3c. Please explain which features were used to create this plot (i.e. did you use principle components of a PCA, selected features or all features)

The features used for this plot are listed in figure EV6. We have listed these features in table EV7.

Figure S22. The pie chart lists 1450 genes, 2188 plasmids. However, the text lists 2281 Plasmids and 1423 genes. Please correct.

Apologies, we corrected this.

Figure S4a. p-value for nuclear pore is missing.

We did not identify any nuclear pore subunits when we profiled cell shape total protein staining features. For this reason, we did not assign a p-value to the 'nuclear pore' node in figure S4A.

Figure S4c. Please specify which features are shown (selected features, best features of the PCA, all features?)

All features used in the cell morphology profiling screen are shown. This has been added to the figure legend of the revised manuscript:

The mean feature values were calculated based on all features used in the cell morphology profiling.

Figure S5. The legend says "Boxplots of the standardized single cell features". However, principal components are listed (mAb414 Texture PC1). Please correct.

Figure S5A indicates boxplots of single cell feature values. The features used include integrated nuclear DAPI intensity, cell shape features, mAb414 staining intensity features and mAb414 staining texture features. To reduce the number of mAb414 texture features we performed PCA on these features and selected the first 30 principal components. For clarity, we have listed these features in table EV7.

Table S2.

Why are some target genes are missing? Please comment on the entry "somethingWrong".

Apologies, the missing entries have been annotated correctly in a revised version of table S2. All targeting plasmids contained in the library are listed. Additional controls (such as the pRG84 empty targeting plasmid) are omitted from the list, explaining the difference between the total library size and the number of entries in table S2.

Table S3.

Each column has two descriptions (first column: 1. "Features morphology profiling:" and 2. "50 principle components of the features:"). Please clarify if all features used for the pca are listed or selected features of the resulting 50 principal components are listed.

All features used for the PCA are listed. To clarify this point we have made the following changes in the revised version of the manuscript:

As a first step in the screen analysis, the dimensionality of the data set was reduced by principal component analysis. The features used for the PCA are listed in Table EV2,3. We selected the first 50 (for the cell morphology profiling) or 30 (for the mAb414 staining features profiling) principal components of the data sets.

The authors state "Our script for selecting gRNA sequences is available on request." If they are willing to distribute it, it should be made available as supplemental material, or run the risk of being unavailable to others' research. Studies show that requesting material "available upon request" has a very low rate of success.

The gRNA selection script and instructions for its use are supplied as a supplementary file: deGroot_gRNAselection.zip.

The language in the title referring to 2 screens is confusing. It is not conventional to describe a single experiment measured for 2 phenotypes as 2 separate screens.

We feel that the use of the plural 'screens' in the title is appropriate because the methods that we describe are suitable to screen any image-based assay and is not limited to the assay that we have described in this manuscript.

In one place, the number of cells per population is described as 10^4 and in two other places as 4×10^3 . Why not just state "~4,000" in each place, to be more precise and consistent?

Apologies, we have made the suggested changes in the revised manuscript.

Check spelling throughout "principle components"

Apologies, we have corrected this in the revised manuscript.

2nd Editorial Decision

21 December 2017

Thank you again for sending us your revised manuscript. We are now satisfied with the modifications made and I am pleased to inform you that your paper has been accepted for publication.

MOLECULAR SYSTEMS BIOLOGY

Corresponding Author Name: Lucas Pelkmans

Manuscript Number: MSB-17-8064